# Designing an International Faculty Development Program in Medical Education: Capacity and Partnership

**Martha Burkle** [1,*]**, Darryl Rolfson** [2] **and Mia Lang** [3]

1   College of Medicine, University of Arizona, Tucson, AZ 85724, USA
2   Office of Undergraduate Medical Education, University of Alberta, Edmonton, AB T6G 2R7, Canada
3   Faculty of Medicine & Dentistry, University of Alberta, Edmonton, AB T6G 2R7, Canada
*   Correspondence: mburkle@arizona.edu

**Abstract:** Providing international medical educators with opportunities for faculty development has become a favorable moment for capacity building and the creation of partnerships with universities around the world. It has also become a social responsibility when such a development implies growth and improvement for the institutions involved. In 2018 and 2019, the University of Alberta Faculty of Medicine & Dentistry designed and delivered an international faculty development program (IFDP) in Edmonton, Canada, in collaboration with the faculty management from Jilin University and Wenzhou Medical University, and Shandong University. The inspiration for program driven by capacity development for three universities in China, all of whom were developing strategies to respond to new government policies for medical education. The focus of the course was based on the needs that the three institutions expressed: teaching innovation, research, and quality curriculum development. By design, the two-week, in-person program included lectures, personal tutorials, class and laboratories observations, as well as guided teaching visits to hospitals and university museums. Recommendations are offered to assist other international faculty development programs focused on capacity building for medical education.

**Keywords:** faculty development; flipped classroom; medical education; evaluation; capacity building; developing countries

## 1. Introduction

This paper presents the analysis and evaluation of an international faculty development program (IFDP) organized and delivered in partnership with three universities from China (Jilin, Shandong, Wenzhou) and the University of Alberta, Canada, in 2018 and 2019 [1].

For context, an overview of the challenges and opportunities facing medical education in China, especially for the local faculty members who were responsible for the program design and delivery, were provided. A "capacity building" framework (Figure 1), was chosen for the provision and support of the faculty development model, to exchange training and 2018nents of the model are presented and analyzed below.

A big challenge for delivery of the program was to share tools and faculty development strategies so that participants could implement these when returning to China to continue their teaching and clinical responsibilities. Participants on both sides were aware of these challenges and possible opportunities implied by their different contexts. These were analyzed continuously bases as the program progressed and reach completion.

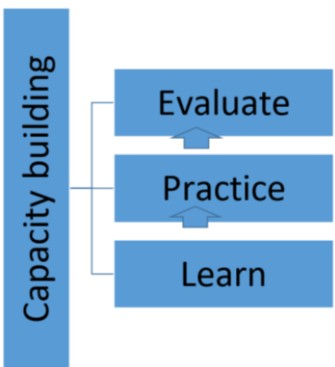

**Figure 1.** Course design for Capacity building: a framework for IFDP [2].

Implementing faculty development methods for capacity building is proposed as one of several strategies to transform medical education in countries as they go through the transformation of health education and health care. Capacity building models for medical education constitute a path to follow to promote excellence in teaching education. Involving the local faculty (at the organizing institution) for course design and development contributed to the idea of "flipping the classroom" as faculty prepared some reading materials before attending the course in person [3].

The 2-week program ended with a full day exercise where participants were encouraged to share with other students (and with the instructors) what it was they learned during the program. Organizers were keen in supporting knowledge transfer and the applications of innovative pedagogies.

### 1.1. Background: Medical Education in China

One of every five doctors in the world works in China [4]. With a population of 1.4 billion, medical services take all forms, from rural doctors (called village doctors) who practice traditional medicine in remote towns and locations, to medical experts who work in large hospitals, providing medical care for up to 10,000 patients per physician.

One factor leading to these wide differences among medical practitioners is the complex medical education in China, resulting from its huge size, lack of standardization, and historical traditions. The duration of medical school training can range from three to eight years. Students enroll in medical school immediately after high school for degrees that span three (diploma), five (bachelor), six (bachelor), seven (master) or eight years (MD).

In 2010, China had 159 medical schools, of which 39 offered a 3-year program and the remainder offered programs lasting five or more years. Almost 1.7 million students filled the classroom space, and more than 400,000 new graduates were reported in 2008 [5,6].

To standardize the quality of medical education the Chinese government launched a major reform in 2009 [7]. Medical schools were directed to increase enrolment of above average students, and to provide better quality teaching that is accredited by the Chinese Medical Doctor Association. After attending a standardized five years of medical school, graduates will continue to a three-year residence program to be eligible for certification in a specialty.

The 2009 medical education reforms also prioritized certain areas for the curriculum and professional certification: quality standardization, curriculum reform, faculty development, and accreditation. The imperative to learn and update skills in these areas gave the faculties of medicine across China the opportunity to grow and develop their skills. This created the opportunity for program designers in other established medical education centers outside of China to use a capacity building framework to target skills development for Chinese medical educators in these prioritized areas.

*1.2. Capacity Building for Medical Education*

Capacity building is a common strategy employed in international development [8,9]. Capacity is defined as "the abilities, behaviors, relationships, and values that enable individuals, groups, and organizations at any level of society to carry out functions or tasks and to achieve their development objectives over time" [8].

In their research on capacity building for medical education in developing countries, Burdick et al. defined successful capacity building as an "increase in the ability of systems to function on their own to meet local needs" [10]. Building capacity is seen as a set of strategies enacted at the level of an entire country, an institution or even an individual in need of growth and development.

Building capacity is not identified as a single intervention, but as a process that starts with an intervention, and continues with evaluation and dialogue over time [11]. Within this chapter, capacity building is applied to the development of knowledge and skills for the improvement of teaching, research, and clinical practice in medical education. A six-step approach to curriculum development [12] was employed to build curricular capacity. In this model, educational development starts with problem identification, local needs assessment, and definition of objectives. This then guides the development of curriculum for teaching and assessment, implementation of the plan, and finally, evaluation of the impact on the program participants.

The four faculties of Medicine mentioned above established a partnership to share knowledge, teaching strategies and resources on faculty development, quality curriculum, and accreditation procedures. From the perspective of capacity building, the agreement was to provide the required skills and training to faculty members in China who are currently teaching at these universities and/or at the associated hospitals.

The provider institution was the University of Alberta (UofA), located in Edmonton, a medium size city in Western Canada. The Faculty of Medicine & Dentistry (FoMD) was established in 1913, and it is one of the world's elite academic health–sciences centres. With 3400 faculty and more than 2000 support staff, the university has one of Canada's largest teaching hospitals where 650 medical students and over 1000 medical residents are being trained at any given time.

Wenzhou Medical University (WMU) is located in the province of Zhejiang, China, with three campuses and 15 Schools. Three hospitals are affiliated to WMU: The First Affiliated Hospital, the Second Affiliated Hospital, and Yuying Children's Hospital. The university has almost 20,000 undergraduate students and 3000 postgraduate students in four faculties (http://en.wmu.edu.cn/About_WMU.htm, accessed on 15 March 2022). WMU requests for faculty development emphasized the undergraduate perspective. WMU managers wanted their faculty to improve teaching and research practices with their undergraduate students.

Shandong University Cheloo College of Medicine (SDU), established in 1902, is one of the earliest universities in China to admit international students for degrees in Bachelor of Medicine and Bachelors of Surgery. SDU has 16 departments, 43,000 undergraduate students, and three affiliated hospitals: Shandong University Qilu Hospital (with 1800 beds, treats 1M patients a year), the Second Hospital of Shandong University (with capacity for 1200 beds), and the Stomatology Hospital for Shandong University (organized into 4 research centres and 2 laboratories) (http://www.mbbs.sdu.edu.cn/, accessed on 7 March 2022).

Jilin University (JLU), or the Norman Bethune Health Sciences Centre is located in Chanchung, China. It is a leading national university under the jurisdiction of China's Ministry of Education and categorized as a Class A Double First Class university. The mandate to become a 5 + 3 degree granted institution was received in 2013. The university was requested to provide leadership and training for the adoption of the new residency model. As in the case of Wenzhou and Shandong universities, medical training at JLU is also associated with two hospitals: Jilin University First Hospital (founded in 1949) with a comprehensive first-class hospital that integrates medical treatment, teaching, scientific

research, health care, and rehabilitation, and Jilin University Second Hospital, that has more than 2789 beds, and more than 4000 faculty members.

*1.3. Needs Assessment for Educational Development*

In order to begin communicating their educational needs, partner institutions (university of Jilin, university of Wenzhou, and Shandong University) hosted a visit with medical education leaders from UofA for two weeks in 2018. The three institutions had different stated educational needs. While Jilin was more focused on developing a RCPSC quality three-year program in Medicine, Surgery and Pediatrics within five years [13], the universities of Wenzhou and Shandong required training on quality undergraduate education [14].

Information received during the visits was complemented by two or more of these events: visits to the FoMD by Chinese university authorities; and email correspondence received from Faculty development leaders from the University of Wenzhou.

The ability of the UofA to design and deliver an effective curriculum plan depended on a robust needs assessment which included other perspectives, apart from the ones mentioned above. However, language and cultural differences made the local needs assessment particularly challenging prior to the in-person delivery of the educational program. These challenges are similar to what Hodges et al. found in their research on global discourse for medical education [15] Some of the needs assessment occurred after the visitors arrived at the UofA, but at this point, the objectives and curriculum design were already largely in place. To design an educational program, organizers and designers needed to better understand the current status in relation to participants' abilities, knowledge, and skills [16]. To overcome this barrier, and further learn about program participants' abilities and knowledge, a more dynamic strategy was added to the program delivery. At the start of each day, there was a quiz or interactive activity to test knowledge acquired from the previous day, and to explore the contents to be examined on that day. Feedback from this activity was then fed forward to incoming instructors.

In this way, the needs assessment was a more continuous process, informed by participatory action methodology [2], in which the program being delivered was adjusted dynamically, based on live participant feedback on their workplace experience. We believe that we were able to better define the desired goals and achievements only after the delivery of the program was completed.

## 2. Faculty Development Methodology

In this section, we will emphasize the methodology and instructional design that led to the educational program delivered to WMU and SDU from 30 September to 11 October 2019. The preceding experiences with JLU were highly informative in the development of this program. From 22 to 26 October 2018, a pilot group of 31 faculty members from JLU, with both clinical and basic sciences backgrounds, attended an in-person academic exchange and medical education program, hosted by the UofA, Faculty of Medicine and Dentistry. Several lessons were learned in terms of logistics, culture, and faculty development were then brought forward in the subsequent planning for the next visit by postgraduate faculty members from JLU in May 2019. A full educational plan for the May 2019 visit was developed including needs assessment, objectives, curriculum, and program evaluation.

A review of the literature on the creation of efficient international partnerships revealed that the IFDP would need to develop a collaborative approach in order to ensure that the program developed would address the needs requested [17]. The IFDP was developed on the bases of expertise knowledge sharing in an environment of mutual respect [18,19]. A participatory faculty development methodology permeated most of the program design and all of the program delivery.

*2.1. IFDP Objectives, Content, and Organization*

An in-person two-week program was designed and delivered to provide knowledge and skills to the participating institutions' faculty. Areas the course focused on were teaching, research, quality curriculum, and accreditation. The course designer created objectives and content based on the needs assessment and requirements from the institutions involved. For the program logistics and organization, a holistic approach to teaching and learning was adopted using formal and informal learning activities [20].

Program objectives:

1. To create a platform for knowledge- transfer and capacity building between the University of Alberta and Wenzhou University, and between the University of Alberta and Shandong University–Cheeloo College of Medicine
2. To build a collaborative partnership between UofA FoMD and Wenzhou Medical University and Shandong University–Cheeloo College of Medicine
3. To share best practices in medical education at the University of Alberta, FoMD
4. To establish UofA Faculty of Medicine & Dentistry as a hub for internationally recognized faculty development programs

Content Development criteria included current trends on teaching innovation, involving students in research activities, and curriculum quality for medical education. Other criteria comprised:

A. Faculty Development contents that were already part of the local faculty development portfolio at the FoMD, UofA
B. The particular learning requirements that were sent to the Faculty development office by the Faculty affairs offices of both Wenzhou and Shandong universities. The Program Coordinator translated these learning requirements into content.
C. Medical education in Canada

A further explanation of these contents is outilined below.

A. A sample of the topics, included in the program, that constituted part of the local faculty development activities were:

  ○ Clinical reasoning
  ○ Assessment
  ○ Problem based learning
  ○ Bedside skills

B. Particular learning requirements received from participant universities:

  ○ Teaching foundations
  ○ Involving students in research activities
  ○ Assessment strategies

C. Medical Education in Canada:

  ○ Canadian Health System
  ○ Quality Teaching at the University of Alberta
  ○ Royal College accreditation system

Other sources of information or criteria that were taken into consideration in the development of the program contents were:

1. The previous IFDP curriculum designed and offered for the cohort in 2018, which used the best practices faculty development model of Steinert [19,21,22]
2. UofA FoMD faculty areas of expertise
3. UofA FoMD faculty who approached the organizers team to offer to teach a particular course within the program.

*2.2. Program Organization and Delivery*

Engaging in faculty development activities to improve teaching and research practices can be considered as one of the many approaches to address complex issues such as the improvement of education in the developing world context [23].

The authors organized the program to be delivered over a two-week period. Included in the program were:

- Lectures and workshops delivered in the classroom (27 in total)
- Meetings (in the form or tutorials) with Academic Chairs and faculty by department of specialty (12)
- Class and labs observations (14)
- Guided visits to Hospitals (Walter C. Mackenzie Health Sciences Centre, Royal Alexandra Hospital)
- Guided visits to UofA FoMD museums and libraries
- Campus tours (including the medical buildings)

All the activities were seen as an opportunity for knowledge transfer and capacity building. Figure 2 below shows the organization of the program by learning activity.

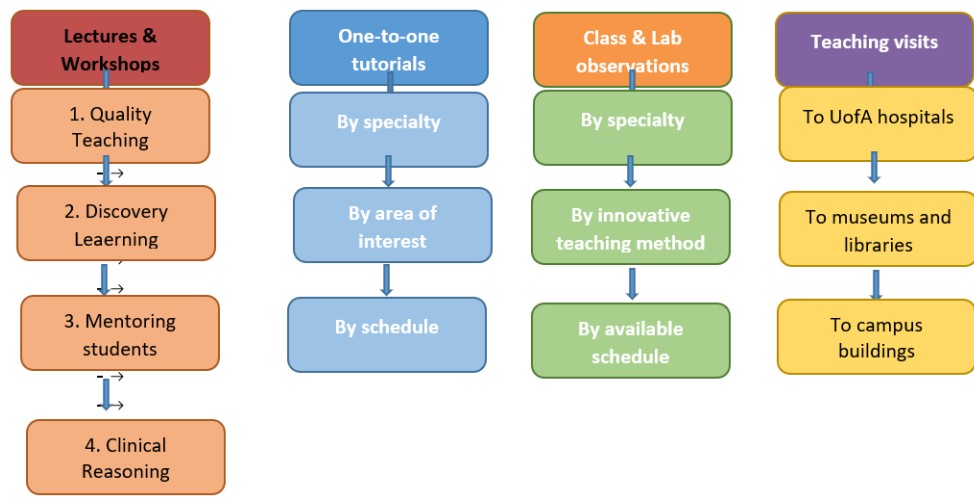

**Figure 2.** International Faculty Development Model [24].

*2.3. The Transformational Change Exercise*

It was important for the program designers that the faculty participants had an opportunity to reflect on what they learned, and how to bring this learning home. A "Transformational change exercise" was included in the last day of the program. Participants were asked to work in teams in the analysis of four areas:

1. Collaborative Learning
2. Changing teaching methods
3. Continuing learning practices
4. New teaching, research, and curriculum development strategies

Thirty-six faculty participated in the exercise. It was important for the program organizers that reflection on these topics was done solely by the team participants. A safe environment was created so that participants could provide feedback.

## 3. Discussion and Further Research

Overall, the IFDP delivery was evaluated very positively both in terms of content and delivery by the participants. During the two weeks that the program lasted, participants requested more one-to-one tutorials with faculty, and opportunities of interaction with students.

Analysis of the data obtained from the transformational change exercise, and from the program evaluation survey, showed that all faculty participating in the program increased

their knowledge regarding the use of innovative technologies. The majority of the course participants felt that their own institutions will benefit from greater involvement of students in research activities.

It always felt that the participants were open to new ideas and methods, and keen to learn and share new ways of learning. Program participants also provided some very helpful ideas on how the UofA could improve their teaching practices. For example, they recommended that we should implement compulsory peer reviewed programs among colleagues, such as the ones they had in China.

Feedback received after the participants returned to their workplaces have made us believe that the IFDP has supported faculty enthusiasm in China to try innovative pedagogies and explore new strategies for curriculum development. However, it is likely that more significant changes may need to be adopted in the future.

Based on the experience of the IFDP, we close this article with some suggestions for future faculty development program delivery in the developing world.

Firstly, it was clear for us that any transformation of teaching initiatives needs to be supported not only by program managers, but also by every single member of the faculty. We acknowledge that our colleagues in China have strict rules and regulations with regard to curriculum development, but it is important to promote and support innovative practices that come from the faculty who work on the front lines with medical students and residents.

Furthermore, we witnessed the principle of the capacity building not being a single intervention but a process [24]. We feel compelled to continue to work with, mentor and tutor, our partners in China, as they explore the use and application of new theories and principles learned while in Canada. We suggest that any faculty development initiative includes a mentorship program that could include, for example, periodic visits from and to the international partner sites.

## 4. Recommendations

Faculty development is a crucial component in medical education. It allows the university and the teaching hospital to remain innovative in their teaching practices. As medicine as a discipline evolves, so does the design, development and implementation of faculty development programs.

In the context of international faculty development curriculum, and after designing, developing, and implementing a program at the University of Alberta in Canada, there are a number of lessons learned to share with the medical education community. These are mentioned below:

### 4.1. Social Accountability Framework

Successful medical universities around the world have a moral commitment (social accountability) for knowledge transfer in the promotion and support of academic excellence and student support. Faculties of Medicine should consider the organization of international programs were knowledge sharing of 'know-how' is the main goal.

### 4.2. Responding to Faculty Needs

To be successful, international faculty development programs should not be planned in isolation, as an inward thinking exercise [25], but should be built as a response to the learning needs of faculty partners. Therefore, in our experience, we confirmed that a successful international faculty development program should not be a rigid and static curriculum, but a flexible one, where changes and approaches could be adjusted according to the local faculty needs.

### 4.3. Sharing, Not Patronizing

As we developed an international faculty development program in partnership with program participants, we soon learned that every medical faculty at a university or at a teaching hospital has its own valuable practices for faculty development and that this

program represented a great opportunity to learn from each other. Addressing the wider global community needs and creating equal opportunities for faculty development [26] should always be part of any international faculty development program.

It is also important to consider the differences that medical education experiments in the different cultural context, and that global homogenous approaches should not be assumed when interacting with global partners [15]

### 4.4. Mentoring, Not Lecturing

A very important component of the IFDP were personal mentoring sessions that the guest faculty had with our local faculty at the UofA. When reviewing the program evaluation, a general opinion of participants was that they learned even more from these one-to-one sessions than from the structured lectures. It was really valuable for all of them to learn about teaching strategies, or supervisory interactions directly from our faculty. We recommend that any faculty development program should include personalized sessions in the form of mentoring or tutoring.

### 4.5. Knowledge Network Creation

A great benefit that both our faculty and the international guests obtained after attending the IFDP was the possibility of building academic partnerships for research and for the exchange of ideas. Commitments were done to continue working in networks for the benefit of students on both sides of the world.

### 4.6. IFDP Program Transformation for Online Delivery

With the international mobility challenges that COVID-19 brought to academic programs around the world, it is important to note here the relevance of transforming the IFDP for online delivery. The program designers and content development team has been working on the transformation of the F2F program into a remote online one. It is still early to analyze the strategies and the challenges to embrace such an important transformation, but we have identified the following steps:

(a) Identification of content that can be re-design for the online environment,
(b) Working with the faculty (original course content designers) for course redesign,
(c) Determining the Learning Management System (LMS) to be used for course creation and delivery,
(d) Building schedules and delivery methods with participating institutions

We recognize that the pandemic brought the opportunity to continue to provide innovative faculty development opportunities and we are sure that, together with our international partners, we will be able to respond to this important challenge.

**Author Contributions:** Conceptualization, M.B.; Formal analysis, D.R.; Project administration, M.L. All authors have read and agreed to the published version of the manuscript.

**Funding:** This research received no external funding.

**Institutional Review Board Statement:** Not applicable.

**Informed Consent Statement:** Not applicable.

**Data Availability Statement:** Not applicable.

**Acknowledgments:** The authors of this chapter would like to recognize the important participation of the participant universities' representatives. Our thanks and recognition go to WMU Chen Hao, JLU Xia Chen, and Shandong University Jinling Yang.

**Conflicts of Interest:** Not applicable.

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
