# Peer review of "Designing an International Faculty Development Program in Medical Education: Capacity and Partnership"

_ime, doi:10.3390/ime2010003_

Round 1

Reviewer 1 Report

L111-140

Please add more information about each university, especially about each school of medicine. And also, you mentioned in the Background that the duration of medical school training in China ranged from 3 years to 8 years, but then only mentioned the study duration of JLU. Please add information on the duration of study at both WMU and SDU.

Additionally, you introduced abbreviations you would be using throughout the piece, but then did not use them often. Please check. 

You have two different figures labeled as Figure 2.

The information for the first Figure 2 seems to be missing. Also, where did this data come from? Please include a source for the data.

L172

There is no description in the Faculty Development methodology.

L198

I recommend to use “3.1” for program objective following “3.2” Program organization and delivery

Fig.2

The second Figure 2 is difficult to understand. What does the arrow indicate?

Does this give an overview? I think there should be an explanation.

L269

What is the breakdown of the 36 participants? Which university? Title?

L270

You mentioned "We created a safe environment..." would you tell me how you did it?

Did you take any surveys from these participants?

It would be more objective if the results of the survey will be added.

L350

If you have started online delivery, it would be wonderful if you explain in detail how you are transitioning.

Author Response

Thank you for your edits and your reviews, Reviewer 1. I have been working on adding your comments and will submit the revised paper this week. 

Kind regards,

Dr. Martha Burkle

Reviewer 2 Report

Congratulations to the authors for implementing an ambitious and complex faculty development program. 

The first line of the paper states that it will "present the analysis and evaluation on an international faculty development program".  It then goes on to describe the program but provides little evidence of analysis or evaluation other than the authors statement of positive results.  In summary, the paper describes how a Canadian school collaborated with three Chinese institutions to develop a program.  It appears that they provided two sessions, one for a single institution more interested in graduate medical education.  It appears that they used this first session to inform the second, focused on UGME.  Unfortunately, they do not share their lessons learned from the first session, just state that it impacted the second.  There are multiple statements in the article that would benefit from some editing and a few typos (e.g., were instead of where).  

My assessment of each section of the article follows:

Introduction - This should contain a clearer depiction of the problem/challenge. Instead they provide some of the methods and results.  Please explain why this is an important collaboration.

Background - the authors provide background for multiple components.  The background about the challenges of medical education is appropriate although it could be strengthened by incorporating some of the challenges based on size that are listed in the section on capacity building.  The background on capacity building (starting on line 83) is excessively long and  provides some of the methodology which belongs elsewhere.  Again, the information on size of the Chinese programs belongs  more in the Background on Medical Education in China.  Part 1.3 could be shortened.  Section 2.0 on Faculty Development is a bit confusing and should, perhaps, include more on the challenges of a cross-cultural program.  Section 2.1 is more about content and hints at what the reader may expect in the analysis and evaluation.  Unfortunately, these data (the results of the Transformational Change Exercise and/or any assessment data from the multiple sessions) are not provided.  It would further strengthen the article to provide any analysis or evaluation by the Canadian participants.  Section 3 - Discussion and further research - provides an overview of the authors analysis without evidence or data.  Seeing some of the actual results - even of qualitative data would change this from "this was good" to somethig with evidence.  Section 4 - Recommendations - Thier recommendations are consistent with what others have found in creating cross-cultural offers but fails to tie their assessment to what is already in the literature.  A quick search revealed at least the following two citations:  

Hodges BD, Maniate JM, Martimianakis MA, Alsuwaidan M, Segouin C. Cracks and crevices: globalization discourse and medical education. Med Teach. 2009 Oct;31(10):910-7. doi: 10.3109/01421590802534932. PMID: 19877863.

Gosselin K, Norris JL, Ho MJ. Beyond homogenization discourse: Reconsidering the cultural consequences of globalized medical education. Med Teach. 2016 Jul;38(7):691-9. doi: 10.3109/0142159X.2015.1105941. Epub 2015 Nov 16. PMID: 26571353.

I am sure there are more.  

Author Response

Dear Reviewer 2 for our paper 

"Designing an International Faculty Development program in Medical Education: Capacity and Partnership"

Thank you for your detailed feedback and your recommendation of adding  important pieces that were originally missing from our work. We have taken the majority of your comments and have added more information, or edited the content so it will make more sense in those sections that you pointed out. One of the references suggested was also added to the original work. 

Thank you again for your review,

Martha Burkle, PhD

Reviewer 3 Report

1.      Abstracts don't normally include citations.

2.      Line 38 Citations needed. Try Aczel, J. C., Peake, S. R., & Hardy, P. (2008). Designing capacity-building in e-learning expertise: Challenges and strategies. Computers & Education, 50(2), 499-510.

a.      “The United Nations Development Programme has defined capacity as “the ability of individuals, organizations and societies to perform functions, solve problems, and set and achieve goals.” (UNDP, 1994). Capacity building in e-learning was given official sanction by the 2005 World Summit on the Information Society, which gave strong encouragement to properly resourced “national strategies for ICT integration in education” (WSIS, 2005).”

                                                                                                               i.     Aczel, Peake, & Hardy, 2008, p.2

3.      “… the instructional design capacity gap needs to be addressed first, followed by the production gap, then the tutorial gap, and finally … attention might be given to community building.”

                                                                                                              i.     ibid, 2008, p.12

4.      Line 58 Do you mean per year?

5.      Line 69 Here, and in many other places, the sentence structure is convoluted. I suggest the authors hire an editor that will make this valuable project more accessible to readers.

6.      Line 97 Rephrase.

7.      Line 104 Repetitive.

8.      Line 111-140 Move to an earlier place in the article.

9.      Line 148-150 Is this needed? If yes, place it somewhere as part of data collection procedures.

10.   Line 277 and other places Check verb usage for appropriateness and consistency.

11.   This is a valuable initiative that warrants knowledge dissemination. However, major revisions in the organization and language usage are needed. Is this a case study? Please describe the participants and the research approach – is this a descriptive case study? An evaluation? Program evaluation is mentioned without reference to a model or method. There is reference to feedback but no findings section.

Author Response

  1. Abstracts don't normally include citations. R. We only used a citation to refer to a previous FD program within the UofA.

  1. Line 38 Citations needed. Try Aczel, J. C., Peake, S. R., & Hardy, P. (2008). Designing capacity-building in e-learning expertise: Challenges and strategies. Computers & Education, 50(2), 499-510. R. Added, thank you

  1. “The United Nations Development Programme has defined capacity as “the ability of individuals, organizations and societies to perform functions, solve problems, and set and achieve goals.” (UNDP, 1994). Capacity building in e-learning was given official sanction by the 2005 World Summit on the Information Society, which gave strong encouragement to properly resourced “national strategies for ICT integration in education” (WSIS, 2005).”
    1. R. This  citation has been added

  1. Aczel, Peake, & Hardy, 2008, p.2

  1. “… the instructional design capacity gap needs to be addressed first, followed by the production gap, then the tutorial gap, and finally … attention might be given to community building.”

  1. ibid, 2008, p.12

  1. Line 58 Do you mean per year? R. Yes, added this information

  1. Line 69 Here, and in many other places, the sentence structure is convoluted. I suggest the authors hire an editor that will make this valuable project more accessible to readers. R. We have edited this section

  1. Line 97 Rephrase. R. Have rephrased. Thank you.

  1. Line 104 Repetitive. 
    1. R. Have edited this line. Thank you

  1. Line 111-140 Move to an earlier place in the article.
    1. R. This comment is not clear since we believe these lines belong to the section that was introduced at this part of the paper.

  1. Line 148-150 Is this needed? If yes, place it somewhere as part of data collection procedures. R. Thank you for this comment, but we believe these words belong here, as part of the methodology related to our work

  1. Line 277 and other places Check verb usage for appropriateness and consistency.

  1. This is a valuable initiative that warrants knowledge dissemination. However, major revisions in the organization and language usage are needed. Is this a case study? Please describe the participants and the research approach – is this a descriptive case study? An evaluation? Program evaluation is mentioned without reference to a model or method. There is reference to feedback but no findings section.

Reviewer 4 Report

An original article. Congratulations to the authors. It is thought that it will make important contributions to the field.

It is recommended that only the minor revisions listed below to be made.

In addition, it should not be done unless it is mandatory from self-citation in the article.

 Delete the names of the institutions and the quotation from the abstract. The abstract should be in a single paragraph.

Author Response

Thank you for your kind words and your review.

As suggested, we have removed the names of the institutions that were cited in the abstract. Also, we have written an extended paragraph since this is the tradition for some journals specialized in Medicine

 Delete the names of the institutions and the quotation from the abstract. The abstract should be in a single paragraph.

Round 2

Reviewer 2 Report

Thank you for sending this back after author revisions. I really appreciate how much work went into the program and into this paper. 

I see some attempts at improving the first paper submitted but find most of my concerns are still present. This isn’t a paper that presents analysis and evaluation. It is a nice description of a program and a commentary by the authors of the benefits of this type of effort with some suggestions for others. It doesn’t really add anything new to the literature in this regard. 

The first line of the paper states that it will "present the analysis and evaluation on an international faculty development program". It still goes on to describe the program with little new evidence added to this version. 

My assessment of each section of the article follows:

·        Introduction - This should contain a clearer depiction of the problem/challenge. Instead, they provide some of the methods and results. Please explain why this is an important collaboration.  The authors have improved the introduction.  It is still long and doesn’t really identify what this paper will add to the literature.

·        Background – The background is somewhat improved but still exceedingly long. 

·        Methods:  There is a lot of description of what was done.  It seems excessive.  They provide the overall program objectives and have added some information about the process/program. 

·        Results:  Unfortunately, data (the results of the Transformational Change Exercise and/or any assessment data from the multiple sessions) are still not provided.  The requested results (evaluation of the components) are still a narrative that essentially says everyone liked it. 

·         Section 3 - Discussion and further research - provides an overview of the authors analysis without evidence or data.  The authors continue to make statements about how participants like it and things are changing but give no examples or even narrative comments to highlight their statements.

·        Section 4 - Recommendations - This section doesn’t appear to have changed much.  My comments that follow are still valid. Their recommendations are consistent with what others have found in creating cross-cultural offers but fails to tie their assessment to what is already in the literature. A quick search revealed at least the following two citations:

·        Hodges BD, Maniate JM, Martimianakis MA, Alsuwaidan M, Segouin C. Cracks and crevices: globalization discourse and medical education. Med Teach. 2009 Oct;31(10):910-7. doi: 10.3109/01421590802534932. PMID: 19877863.

·        Gosselin K, Norris JL, Ho MJ. Beyond homogenization discourse: Reconsidering the cultural consequences of globalized medical education. Med Teach. 2016 Jul;38(7):691-9. doi: 10.3109/0142159X.2015.1105941. Epub 2015 Nov 16. PMID: 26571353.

I am sure there are more.

Author Response

Dear reviewer:

We have taken very seriously your recommendations and have edited our paper following them in a very detailed form. Below are the changes we have done to the paper: 

1. Introduction - This should contain a clearer depiction of the problem/challenge. We have added this.

2. Background – The background is somewhat improved but still exceedingly long. We have edited this section to decrease its size. Kindly note that other reviewers mentioned we needed more info on the university participants

3. Methods:  There is a lot of description of what was done.  It seems excessive. We have edited this section

4. Results:  Unfortunately, data (the results of the Transformational Change Exercise and/or any assessment data from the multiple sessions) are still not provided.  The requested results (evaluation of the components) are still a narrative that essentially says everyone liked it. Kindly note that this exercise included only qualitative data. This is why what we shared in the paper did not provide any stats. 

Discussion and further research - provides an overview of the authors analysis without evidence or data.  We have edited this section and included more of the qualitative data provided by the participants

Section 4 - Recommendations - This section doesn’t appear to have changed much.  We have added a discussion on the two citations you mentioned. Thank you for this. WE believe recommendations are now more solid.

Reviewer 4 Report

Yes, I accept. 

Thanks alot.